# Vancomycin Elution Kinetics of Four Antibiotic Carriers Used in Orthopaedic Surgery: In Vitro Study over 42 Days

**DOI:** 10.3390/antibiotics12111636

**Published:** 2023-11-17

**Authors:** Maria Anna Smolle, Hana Murtezai, Tobias Niedrist, Florian Amerstorfer, Nina Hörlesberger, Lukas Leitner, Sebastian Martin Klim, Reingard Glehr, Raju Ahluwalia, Andreas Leithner, Mathias Glehr

**Affiliations:** 1Department of Orthopaedics and Trauma, Medical University of Graz, 8036 Graz, Austria; 2Institute of Pharmaceutical Sciences, University of Graz, 8010 Graz, Austria; 3Clinical Institute of Medical and Chemical Laboratory Diagnostics, Medical University of Graz, 8036 Graz, Austria; 4Institute of General Practice and Evidence-Based Health Services Research, Medical University of Graz, 8036 Graz, Austria; 5Orthopaedics, Kings College Hospital NHS Foundation Trust, London SE5 9RS, UK

**Keywords:** biodegradable antibiotic carrier, elution kinetics, vancomycin

## Abstract

This study aimed to analyse and compare the vancomycin elution kinetics of four biodegradable, osteoconductive antibiotic carriers used in clinical practice within a 42-day in vitro setting. Carriers A and D already contained vancomycin (1.1 g and 0.247 g), whereas carriers B and C were mixed with vancomycin according to the manufacturer’s recommendations (B: 0.83 g and C: 0.305 g). At nine time points, 50% (4.5 mL) of the elution sample was removed and substituted with the same amount of PBS. Probes were analysed with a kinetic microparticle immunoassay. Time-dependent changes in vancomycin concentrations for each carrier and differences between carriers were analysed. Mean initial antibiotic levels were highest for carrier A (37.5 mg/mL) and lowest for carrier B (5.4 mg/mL). We observed time-dependent, strongly negative linear elution kinetics for carriers A (−0.835; *p* < 0.001), C (−0.793; *p* < 0.001), and D (−0.853; *p* < 0.001). Vancomycin concentrations increased from 48 h to 7 d and dropped thereafter in carriers C and D whilst constantly decreasing at any time point for carrier A. Carrier B showed a shallower decrease. Mean antibiotics levels at 42 d were 1.5 mg/mL, 2.6 mg/mL, 0.1 mg/mL, and 0.1 mg/mL for carriers A, B, C, and D. Differences in mean initial and final vancomycin concentrations for carrier A were significantly larger in comparison to C (*p* = 0.040). A carrier consisting of allogenic bone chips showed the highest vancomycin-to-carrier ratio and the largest elution over the study period. Whilst vancomycin concentrations were still measurable at 42 days for all carriers, carrier A provided a higher drug-to-carrier ratio and a more consistent antibiotic-releasing profile.

## 1. Introduction

Large bone defects resulting from chronic osteomyelitis, implant-associated infections, or extensive trauma and tumour resections pose a significant problem in orthopaedic and trauma surgery. To enhance mechanical stability, reduce dead space, and create novel bone stock, these defects may be filled with autografts, allografts, or synthetic devices [1,2,3,4].

Despite their favourable biological properties, associated donor-site morbidity and limited availability have to be seen as major disadvantages of bony autografts over allografts [1,5]. However, allografts may be rejected by the recipient or lose their osteogenic properties during processing [1,6]. Notably, the risk of disease transmission from allografts has been minimised with the introduction of thorough screening and novel processing methods [7,8,9].

Currently, resorbable synthetic devices, such as calcium sulphate-, hydroxyapatite-, and tricalcium phosphate-based materials, and non-resorbable synthetic devices like polymethylmethacrylate (PMMA), are also frequently used in clinical practice [10,11,12,13]. Biodegradable synthetic bone fillers have been refined to act as antibiotic carriers and not only serve as a local scaffold [13,14,15,16,17]. On the other hand, the high primary mechanical stability and low costs of non-resorbable synthetic bone fillers can be seen as advantageous over resorbable ones [18]. Furthermore, whilst non-resorbable devices will lose their ability to release antibiotics at some point in time [17,19,20] and thereafter may be colonised with remaining bacteria again, forming an antibiotic-resistant biofilm [21,22], resorbable bone fillers are ultimately replaced by host bone.

An essential property of an antibiotic carrier is the release of its drug over a prolonged period, whilst antibiotic concentrations at subinhibitory levels have to be avoided, as they may promote resistant pathogens [23,24]. Currently, there are only a few studies comparing more than one biodegradable antibiotic carrier within the same setting regarding antibiotic-releasing properties and/or efficacy [15,24,25]. This study aimed primarily at comparing the vancomycin elution kinetics of four biodegradable, osteoconductive antibiotic carriers used in routine clinical practice. Secondarily, elution was analysed throughout 42 consecutive days within the same in vitro setting to determine the longevity of release.

## 2. Results

### 2.1. Mean Concentration Differences between Antibiotic Carriers

Mean initial vancomycin concentrations (4 h) were highest in carrier A (37.5 mg/mL), whilst the lowest values were found for carrier B (5.4 mg/mL). In between were the mean initial vancomycin concentrations of carrier C (6.4 mg/mL) and D (7.3 mg/mL). Vancomycin concentrations of carriers A and B did not fall below 1.5 mg/mL during the 42 days, but concentrations of carriers C and D reduced to <1.5 mg/mL at day 21 (Figure 1).

At the last measurement, 42 days after study setup, mean vancomycin concentrations were 1.5 mg/mL, 2.6 mg/mL, 0.1 mg/mL, and 0.1 mg/mL for carriers A, B, C, and D, respectively (Table 1). When considering the mean difference in antibiotic levels from the first to the last measurement, a significantly larger difference was present for carrier A in comparison to C (*p* = 0.040). However, mean vancomycin concentration differences from the first to last measurement were nonsignificant between carriers A and B (*p* = 0.261), A and D (*p* = 0.152), B and C (*p* = 0.134), B and D (*p* = 0.350), and C and D (*p* = 0.236).

### 2.2. Time-Dependent Change in Individual Antibiotic Carriers

The overall results showed that Pearson’s correlation coefficient was −0.835 for carrier A (*p* < 0.001), −0.290 for carrier B (*p* = 0.087), −0.793 for carrier C (*p* < 0.001), and −0.853 for D (*p* < 0.001). Both carriers A and B exhibited their highest mean concentrations at the start of the study, whilst carriers C and D showed a small rise in antibiotic concentration from 48 h to day 7 and thereafter consistently decreased. The individual carrier results are described below:(a)Carrier A: A statistically significant difference in mean antibiotic concentrations over time was present (*p* < 0.001; Table 1). Although an overall steady decrease in antibiotic concentration was observed during the study period, there was no difference between any nearest time points (all *p* > 0.05).(b)Carrier B: Neither a significant difference in mean vancomycin concentration over time (*p* = 0.921) nor between nearest time points was observed (all *p* > 0.5; Table 1). Notably, a very shallow decrease in vancomycin concentration over time was present, starting from 4.5 mg/mL at 48 h and reaching 2.6 mg/mL at 42 days.(c)Carrier C: There was a significant mean difference in antibiotic concentration over the study period (*p* < 0.001; Table 1). However, apart from a significant drop in vancomycin concentration from day 7 to day 14 by 5.9 mg/mL (*p* < 0.001), no difference between the nearest time points was observed (*p* > 0.05).(d)Carrier D: Significant mean differences in vancomycin concentrations over time were observed (*p* < 0.001; Table 1). Assessing nearest time points, a significant decrease in vancomycin concentration between 8 and 24 h (−1.5 mg/mL; *p* = 0.001); a significant increase between 8 and 24 h (+5.6 mg/mL; *p* < 0.001); and subsequently significant decreases between 24 and 48 h (−2.1 mg/mL; *p* < 0.001), 48 h and 7 days (−3.9 mg/mL; *p* < 0.001), 7 and 14 days (−3.0; *p* < 0.001), as well as 14 and 21 days (−1.3; *p* = 0.003) were observed (Table 1).

## 3. Discussion

In the current experimental study, the highest initial concentrations of vancomycin were measured for carrier A, which is composed of human bone chips. Concentrations decreased steadily and were more pronounced in human bone allograft carrier A than in biodegradable carriers C and D. Furthermore, at the last measurement after 42 days, vancomycin levels were highest in the elution assay of carrier A. Notably, for carrier B, composed of human bone chips manually mixed with vancomycin, the decrease in vancomycin concentrations was not statistically significant, with initially moderate, and thereafter constantly low levels.

In the clinical setting of bone defects and infection, the delivery of local antibiotics at high concentrations is warranted to effectively target pathogens while minimising systemic side effects [24]. Moreover, rather than a single peak in antibiotic concentrations, a constantly high concentration over a prolonged period is necessary to attack bacteria and avoid the development of antibiotic resistance [26]. This can be achieved through antibiotic carriers based on autologous or allogenic bone, biodegradable bone substitutes, or non-degradable synthetic devices. Notably, antibiotic elution kinetics significantly vary depending on the type of carrier, its preparation, as well as on the ratio of carrier to the drug [24]. All these factors bear the potential to eventually affect the carriers’ efficacy in clinical practice.

The current study observed high initial vancomycin concentrations for carrier A (allograft bone chips) that constantly decreased thereafter. Vancomycin elution kinetics for carrier B, consisting of pre-infused human allograft bone chips, were lowest at the beginning and showed an overall shallow decrease. An explanation for the inconsistent vancomycin release in carrier B could be the differing vancomycin concentration per carrier, being higher in carrier A (1.1 g vancomycin per sample) than in carrier B (0.83 g per sample). On the other hand, it may also be related to varying impregnation techniques, with carrier B being manually impregnated with a vancomycin solution, whilst carrier A was ready for use as provided by the manufacturer. Even so, both biodegradable synthetic bone carriers had even lower vancomycin concentrations (carrier C: 0.305 g per sample; carrier D: 0.247 g per sample) than carriers A and B.

Differences in the chemical composition of the four antibiotic carriers tested may likewise have influenced the elution kinetics; other than calcium sulphate-based bone void fillers (as carrier C), those composed of tricalcium phosphate and hydroxyapatite only require degradation by osteoclasts and macrophages as they are stable at physiologic pH [1,27]. This results in dissolution times between 6 and 24 months and from 24 to 60 months from the implantation for tricalcium phosphate- and hydroxyapatite-based bone void fillers, respectively [28,29,30,31]. On the other hand, calcium sulphate-based carriers are resorbed between 6 and 12 weeks from implantation in the human body [32,33]. By combining calcium sulphate with tricalcium phosphate or hydroxyapatite (as carrier D in the present study), porosity and decomposition can be improved, as the resorbed calcium sulphate upon implantation creates channels in the carrier, allowing for later bony ingrowth [1]. In comparison, the specifically prepared allograft bone void fillers herein investigated, carriers A and B, are composed of decellularized bone matrix containing collagen, osteoinductive proteins (to some extent), and minerals that promote osteoconduction [34]. This results in a faster and more complete bony integration [34,35]. However, allografts are usually not completely resorbed by host osteoclasts, with remnants of the allograft still present years after implantation [36]. Given these chemical composition differences between allograft and synthetic bone void fillers, the variations in vancomycin elution kinetics observed herein are unsurprising.

They are in accordance with those reported by Kucera et al. [24], who assessed allogenic cancellous human bones and tricalcium phosphate, amongst others, as carriers for vancomycin and gentamycin [24]. Higher initial vancomycin concentrations were measured for one allogenic bone carrier in comparison to the tricalcium phosphate synthetic carriers [24]. However, Kucera et al. were unable to measure any antibiotic concentrations from 17 days onward in their elution assays, whilst we detected antibiotics in the sample up to day 42 [24]. A possible explanation for the differences is in line with other investigators [24,37] who also used allograft bone as antibiotic carriers but exchanged the entire elution medium at different sampling time points, thus eventually leading to unquantifiable vancomycin levels after several rounds [24,37]. This hypothesis is supported by the findings from Bormann et al., reporting on Staphylococcus aureus growth inhibition upon treatment with elution media deriving from assays of vancomycin-carrying demineralised bone matrix up to day 56. Their experimental setup exchanged only 50% of the elution media at each time point [38]. We believe that the approach to substitute the removed elution media with the same amount of PBS may more accurately depict and mimic in vivo situations where (except for iatrogenic interventions) moderate but constant fluid exchange can be expected.

High vancomycin concentrations observed in the elution assay for carrier A may have a negative effect on local osteoblasts required for the revitalisation of the “dead space” filled with either allogenic bone or biodegradable synthetic devices [24]. However, it has been shown that vancomycin concentrations up to 20 mg/mL cause only a minor decrease in DNA content and alkaline phosphatase activity of osteoblasts [39] and do not significantly impair mesenchymal stem cell proliferation rate [24]. Our observed values are lower than this rate.

The maximum antibiotic concentrations reached with PMMA upon surgery, even during the first few days, are generally lower (median: 0.5 mg/mL vancomycin in drainage fluid) than those observed for the current biodegradable antibiotic carriers in vitro [40]. This is mainly caused by the fact that specific drug-to-PMMA ratios should be adhered to in order to maintain mechanical stability [41]. For example, per 40 g of PMMA, no more than 0.5 mg of vancomycin should be added [41], equivalent to a vancomycin concentration of 0.013 g per 1 g bone cement. On the other hand, the herein-used antibiotic carriers contained—as suggested by the manufacturers—vancomycin concentrations from 0.045 g to 0.55 g per 1 g of carrier.

Some limitations have to be taken into consideration when interpreting the results. First, the exclusive focus on vancomycin elution kinetics rather than additionally performing experiments such as the zone of inhibition testing, monitoring of cytotoxicity, and efficacy analysis that may have provided further insights into the behaviour of the four carriers poses a potential limitation. Second, by not substituting the entire elution medium at every point in time but rather substituting 50% of it, the “wash-out” of antibiotics may have proceeded at a different rate than in in vivo. On the other hand, the current setup may mimic in vivo processes more accurately through partial substitution of surrounding fluids with time than in experiments in which the entire elution medium is constantly exchanged [24,37]. Third, variations in chemical composition and drug-loading abilities of the four carriers tested may have potentially influenced antibiotic release with time. Further, vancomycin carrier concentrations, as recommended by the manufacturer, were used for the preparation of samples, naturally leading to differing amounts of vancomycin in each carrier. Yet, it could be ensured that the experimental setup followed the clinical protocol, given that the aim of this study was to allow a direct comparison of vancomycin-releasing properties of antibiotic carriers routinely used in clinical practice within a similar reproducible environment. Further studies are required to assess any effect on bone cells and assess if these results can be replicated in clinical endpoints of interest, such as re-infection, in in vitro studies.

## 4. Materials and Methods

In this in vitro study, four different antibiotic carriers used in orthopaedic and trauma surgery were tested for their releasing properties of vancomycin, namely (A) *OSmycin V^®^* (European Cell and Tissue Bank, Wels, Austria), (B) *Ospure™* (European Cell and Tissue Bank, Wels, Austria), (C) *STIMULAN^®^* (Biocomposites, Keele, UK), and (D) *CERAMENT™V* (Bone Support, Lund, Sweden). To simulate an in vivo situation, the respective carriers were prepared as recommended by the manufacturers. Notably, as volume is the limiting factor upon in vivo application, we used volume rather than weight as a quantitative measure for each carrier, which was more akin to the clinical setting.

### 4.1. Description of Antibiotic Carriers

(a)Carrier A (*OSmycin V^®^* (European Cell and Tissue Bank, Wels, Austria)) consists of decellularized and delipidated human allograft bone chips, sized 1.5–10 mm, that have been impregnated with 0.56 g vancomycin per 1 g of bone chips.(b)Carrier B (*Ospure™* (European Cell and Tissue Bank, Wels, Austria)) likewise contains delipidated and decellularized human allograft bone chips without any additional substances.(c)Carrier C (*STIMULAN^®^* (Biocomposites, Keele, UK)) is a calcium-sulphate matrix that can be mixed with tobramycin, gentamycin, and vancomycin. Prior to implantation into bone, the carrier is shaped into small beads, which promptly solidify. For the current study, 10 mL of carrier C was mixed with 1 g of vancomycin, as suggested by the manufacturer.(d)Carrier D (*CERAMENT™V* (Bone Support, Lund, Sweden)) constitutes a vancomycin-loaded hydroxyapatite/calcium-sulphate composite that can either be injected directly into the bone or formed into beads prior to implantation [26]. It contains 0.66 g vancomycin per 10 mL of carrier.

### 4.2. Preparation of Antibiotic Carriers

Carriers A and D were ready for use as they already contained vancomycin, whereas carriers B and C had to be impregnated with vancomycin (as recommended by the manufacturers’ guidelines). Due to differences in the densities among the individual carriers, a volume of 5 mL, rather than a specific weight, was determined for each sample. Therefore, a typical clinical use situation could be prepared for comparison purposes. To achieve the aforementioned volume, 2.0 g of carrier A, 1.5 g of carrier B, 5.0 g of carrier C, and 5.5 g of carrier D (Table 2) were used. All preparations were performed in a laminar flow workbench to minimise contamination risk. Four samples from each carrier were set up.

(a)For the preparation of carrier A, four wells of a six-well plate were each filled with 5 mL of the carrier as provided by the manufacturer (equivalent to 2 g, containing 1.1 g of vancomycin each) and pressed with a 35 lb-calibrated indenter (Figure 2).

(b)Firstly, 4 wells of a 6-well plate (Figure 2) were filled with 5 mL of carrier B and subsequently impregnated with a physiological saline solution-dissolved vancomycin (Vancomycin Hikma, Hikma Pharmaceuticals, Sintra, Portugal) for 15 min at room temperature. This allowed the bone chips to absorb the vancomycin solution. The resulting samples were equivalent to 1.5 g containing 0.83 g of vancomycin each and were ultimately pressed with a 35 lb-calibrated indenter.(c)For carrier C, 1 g of vancomycin powder was first mixed with 10 mL of the solid phase of the carrier. Thereafter, 6 mL of the mixing solution provided by the manufacturer was added and thoroughly mixed for 30 s. Within 1–2 min of combining the liquid and solid phases, the mixture was transferred to the bead mat provided by the manufacturer, with a bead size of 6 mm. The mat was gently tapped on the table to remove any potential air bubbles. After another 8 min of setting time, the beads were released from the bead mat by bending it back and forth. Again, four wells of a 6-well plate were each filled with 5 mL of beads (containing 0.305 g of vancomycin) and pressed with the indenter calibrated to 35 lbs (Figure 2).(d)Carrier D was prepared according to the package insert. First, 1 g of vancomycin powder was dissolved in an iodine-based liquid phase (iohexol, provided by the manufacturer in a syringe). The mixture was subsequently transferred to the mixing device containing hydroxyapatite and calcium sulphate hemihydrate powders. The liquid and solid phase were subsequently combined for 30s using the manufacturer’s mixing device, equivalent to one complete turn per second. Thereafter, 5 mL of the prepared carrier was inserted into 4 wells of a 6-well plate each (containing 0.247 g of vancomycin) and allowed to settle for 3 min. Afterwards, probes were pressed with a 35 lb-calibrated indenter (Figure 2).

### 4.3. In Vitro Elution Assay

Releasing profiles of the four vancomycin carriers were determined using an in vitro elution method. Phosphate-buffered saline (PBS, pH 7.4) was used as a dissolution medium. Each sample containing 5 mL of the antibiotic carrier was mixed with 9 mL of PBS and incubated at 37 °C.

Once per week, and after the transfer of the 6-well plates to the laminar workbench, the incubator and water bath were cleaned with Bacillol^®^ tissues under sterile conditions. Additionally, the water bath was exchanged using 500 mL of distilled water mixed with 2.5 mL of AquaClean (antimicrobial additive consisting of ammonium compounds) [42]. Thereafter, the new water bath and 6-well plates were retransferred to the incubator.

Antibiotic release was assessed over 42 consecutive days, with samples collected at 9 distinct points in time (4 h, 8 h, 24 h, 48 h, 7 days, 14 days, 21 days, 35 days, and 42 days). At every time point, 50% of the elution medium (i.e., 4.5 mL) was removed from each sample and substituted with 4.5 mL of PBS [38]. Prior to sampling, the medium of each well was mixed with a 1000 µL single-channel micropipette to avoid falsified fluctuations in the concentration of the antibiotic. Thereafter, 1.5 mL of the elution medium was removed using a 100–1000 µL single-channel micropipette (procedure: 3 × 500 µL) and transferred to Eppendorf tubes. Further, 3.0 mL of the elution medium was removed (procedure: 3 × 1000 µL) and discarded. The “forward pipetting” technique (i.e., the pipette’s control knob is pushed down to the first pressure point before the pipette’s tip is dipped into the elution medium, and the control knob is released for elution medium uptake) was employed. A total of 4.5 mL of removed elution media was substituted with the same amount (4.5 mL) of PBS (procedure: 4 × 1000 µL + 1 × 500 µL).

Samples were stored in Eppendorf tubes at −20° until laboratory analysis.

### 4.4. Kinetic Microparticle Immunoassay

Vancomycin levels in the samples were determined using a kinetic microparticle immunoassay (ONLINE TDM Vancomycin Gen.3, Roche Diagnostics, Vienna, Austria) applied on a cobas^®^ 8000 c502 analyzer (Roche Diagnostics, Rotkreuz, Switzerland). This method is routinely used for vancomycin measurements in blood serum and has been standardised against USP reference standards.

To assess the efficacy of the test and provide robust control, as incomparable sample materials were used in this study and the vancomycin concentrations were expected to be much higher in the study samples than in serum, the method had to be validated for its use in this extraordinary sample matrix. Several standard additional assays were performed with different concentrations of pure vancomycin (Vancomycin Hikma, Pfizer Corporation Austria Gesellschaft m.b.H., Vienna, Austria) and the buffer solution, and results were measured in repeated runs with several dilution media (distilled water, isotonic saline solution, and serum albumin). The results from the pre-study validation revealed a recovery of nearly 100% when the samples were diluted with serum albumin (Alburex 5%, CSL Behring, King of Prussia, PA, USA). A 1000-fold dilution was necessary to achieve linear results. The within-run precision was 6% for a vancomycin concentration of roughly 46 mg/mL. Based on the results from the method validation study, each sample from the elution assay was diluted at least 1000-fold with serum albumin prior to measurement with the kinetic microparticle immunoassay.

### 4.5. Statistical Methods

Mean values and standard deviations of the antibiotic levels were calculated at each respective time point from the start of the assessment to day 42. Pearson’s correlation coefficient was used to test for a linear relationship of elution kinetics for each carrier over time, with an absolute value of R > ±0.7 considered a strong correlation. One-way repeated measures analysis of variance (ANOVA) was applied to assess overall changes in mean antibiotic concentrations over time. Subsequent pairwise comparison of means, adjusting for multiple comparisons with the Tukey method, was performed to assess differences in antibiotic concentrations between consecutive time points for each carrier. Kruskal–Wallis test with post hoc Dunn test was applied to assess significant differences between carriers’ antibiotic concentrations from the first (4 h) to the last measurement (42 days).

A two-sided *p*-value of <0.05 was considered statistically significant.

## 5. Conclusions

Using an in vitro model to assess elution characteristics of four commonly prescribed antibiotic carriers, pre-prepared allogenic bone chips exhibited the highest drug-to-carrier ratio, highest initial antibiotic release, and a constant favourable releasing profile, which remained higher than that of all non-allogenic biodegradable carriers tested for up to 42 days.

## Figures and Tables

**Figure 1 antibiotics-12-01636-f001:**
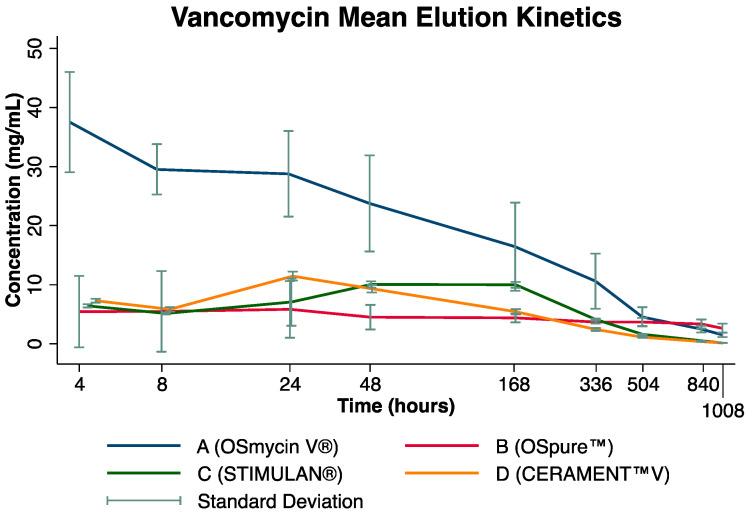
Elution kinetics of mean vancomycin concentration (in mg/mL) for the four antibiotic carriers over time (depicted on a logarithmic scale for better visualisation).

**Figure 2 antibiotics-12-01636-f002:**
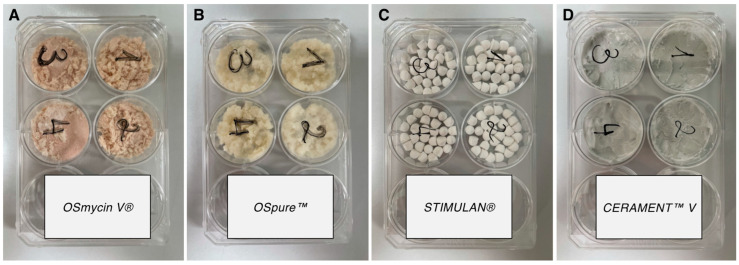
Experimental setup depicting four 6-well plates, each with 4 wells filled with the prepared antibiotic carriers in the elution medium: (**A**) carrier A (*OSmycin V^®^*), (**B**) carrier B (*OSpure™*), (**C**) carrier C (*STIMULAN^®^*), and (**D**) carrier D (*CERAMENT™ V*)).

**Table 1 antibiotics-12-01636-t001:** Mean vancomycin concentration (in mg/mL) for the four antibiotic carriers at specific time points (also see Figure 1). One-way repeated measures analysis of variance (ANOVA) was performed for each carrier.

Time	4 h	8 h	24 h	48 h	7 d	14 d	21 d	35 d	42 d	Overall Difference (ANOVA)
Carrier A (*OSmycin V^®^*)	37.5 ± 8.5	29.5 ± 4.3	28.8 ± 7.3	23.8 ± 8.1	16.4 ± 7.5	10.6 ± 5.7	4.5 ± 1.6	2.5 ± 0.6	1.5 ± 0.4	F(8, 24) = 54.0 *p* < 0.001
Carrier B (*OSpure™*)	5.5 ± 6.1	5.5 ± 6.8	5.8 ± 4.8	4.5 ± 2.1	4.4 ± 0.7	3.7 ± 0.2	3.7 ± 0.7	3.3 ± 0.8	2.6 ± 0.8	F(8, 24) = 0.55 *p* = 0.921
Carrier C (*STIMULAN*)	6.4 ± 0.3	5.1 ± 0.2	7.1 ± 4.0	10.0 ± 0.5	10.0 ± 0.5	4.1 ± 0.2	1.6 ± 0.1	0.5 ± 0.1	0.1 ± 0.05	F(8, 24) = 29.8 *p* < 0.001
Carrier D (*CERAMENT™V*)	7.3 ± 0.3	5.8 ± 0.4	11.5 ± 0.8	9.3 ± 0.6	5.4 ± 0.4	2.4 ± 0.2	1.1 ± 0.1	0.4 ± 0.05	0.1 ± 0.01	F(8, 24) = 521.8 *p* < 0.001

**Table 2 antibiotics-12-01636-t002:** Details about carriers, weight per sample, and vancomycin concentration.

	Carrier A	Carrier B	Carrier C	Carrier D
**Product**	*OSmycin V^®^*	*OSpure™*	*STIMULAN^®^*	*CERAMENT™V*
**Volume per sample**	5 mL	5 mL	5 mL	5 mL
**Weight per sample**	2.0 g	1.5 g	5.0 g	5.5 g
**Vancomycin per sample**	1.1 g	0.83 g	0.305 g	0.247 g
**Vancomycin per 1 g carrier**	0.55 g	0.55 g	0.06 g	0.045 g

## Data Availability

Data are available upon reasonable request from the corresponding author.

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
