# Peer review of "Vancomycin Elution Kinetics of Four Antibiotic Carriers Used in Orthopaedic Surgery: In Vitro Study over 42 Days"

_antibiotics, 2023, doi:10.3390/antibiotics12111636_

Round 1

Reviewer 1 Report

Comments and Suggestions for Authors

The manuscript entitled "Vancomycin elution kinetics of four antibiotic carriers used in 2 orthopaedic surgery. In-vitro study over 42 days."represents work based on vancomycin and it different carriers. 

The author must enhance the work after giving some more justification for the result observation based on the carrier's composition. Is there any difference because of the chemical composition? I feel in the present form the paper is not much clear for the non-clinical population. 

Comments on the Quality of English Language

seems ok 

Author Response

Comment: The manuscript entitled "Vancomycin elution kinetics of four antibiotic carriers used in 2 orthopaedic surgery. In-vitro study over 42 days."represents work based on vancomycin and it different carriers.

The author must enhance the work after giving some more justification for the result observation based on the carrier's composition. Is there any difference because of the chemical composition? I feel in the present form the paper is not much clear for the non-clinical population.

Author Response: The authors would like to thank the reviewer for their time and the provided improvement suggestion with regards to differences in chemical composition of each antibiotic carrier. An additional paragraph has been added to the discussion section of the manuscript, now highlighting variations in chemical composition of the individual carriers, and the resulting differing properties (e.g. time to resorption, bony integration; see revised manuscript; Discussion, page 5, lines 149-167).

Furthermore, this issue has been added as potential limitation to the study that should be taken into consideration when interpreting the results (see revised manuscript; Discussion, page 6, lines 210-211).

Reviewer 2 Report

Comments and Suggestions for Authors

This study analyzed and compared the vancomycin elution kinetics of four biodegradable antibiotic carriers used in clinical practice over a 42-day in-vitro period. Carriers A and D contained initial vancomycin levels of 1.1 g and 0.247 g, while carriers B and C were mixed with vancomycin as per the manufacturer's recommendations, with initial levels of 0.83 g and 0.305 g, respectively. The study found that vancomycin concentrations decreased over time for all carriers, with carrier A having the highest initial levels and a more consistent antibiotic-releasing profile compared to carrier C. My major concern is, the initial concentrations of four kinds of carriers, and the drug loading ability are all different, so I'm not sure about the scientificity of the results.

Author Response

Comment: This study analyzed and compared the vancomycin elution kinetics of four biodegradable antibiotic carriers used in clinical practice over a 42-day in-vitro period. Carriers A and D contained initial vancomycin levels of 1.1 g and 0.247 g, while carriers B and C were mixed with vancomycin as per the manufacturer's recommendations, with initial levels of 0.83 g and 0.305 g, respectively. The study found that vancomycin concentrations decreased over time for all carriers, with carrier A having the highest initial levels and a more consistent antibiotic-releasing profile compared to carrier C. My major concern is, the initial concentrations of four kinds of carriers, and the drug loading ability are all different, so I'm not sure about the scientificity of the results.

Author Response: The authors would like to thank the reviewer for this critical appraisal of the manuscript and the comments provided regarding the 1) differences in initial concentrations of the four carriers, and the 2) potentially differing drug loading ability.

Regarding point 1), the initial vancomycin concentrations were indeed different, owing to recommendations provided by the manufacturers of the individual antibiotic carriers. This allowed for simulation of a real-life clinical setting, but also resulted in the aforementioned differences in initial antibiotic concentrations per carrier. This issue is now outlined in the limitations section of the manuscript (see revised manuscript; Discussion, page 6, lines 211-217).

Regarding point 2), the authors would like to emphasize that the research rationale of this study was to compare different antibiotic carriers that most likely present with varying antibiotic loading- and release-properties. As differences in the chemical composition of each carrier may have indeed influenced vancomycin releasing properties, a novel section has been added to the discussion of the manuscript, shedding light on varying properties of the investigated carriers owing to their composition (see revised manuscript; Discussion, page 5, lines 149-167). This point has also been added to the limitations section of the manuscript (seerevised manuscript; Discussion, page 6, lines 210-211).

Reviewer 3 Report

Comments and Suggestions for Authors

The manuscript presently focuses on the vancomycin elution kinetics from various carriers. It would be beneficial for the scope of the study to be expanded to include more comprehensive in vitro and ex vivo investigations to underscore its relevance. As it stands, the narrow focus on elution kinetics without additional experimental contexts limits the significance of the work, which may not meet the publication criteria for antibiotics.

Regarding Figure 1, it is recommended that the author employ a logarithmic scale for the time axis to enhance clarity and prevent the overlapping of time points, particularly at 0 and 8 hours.

In Section 4.3, pertaining to the in vitro elution assay, the methodology for sample analysis needs to be delineated clearly. The statement "a previously reported methodology" is insufficient. The method of analyzing vancomycin concentration is critical for the credibility of the manuscript, and readers would benefit from a detailed description of the analytical technique employed.

Comments on the Quality of English Language

Minor editing of English language required

Author Response

Comment: The manuscript presently focuses on the vancomycin elution kinetics from various carriers. It would be beneficial for the scope of the study to be expanded to include more comprehensive in vitro and ex vivo investigations to underscore its relevance. As it stands, the narrow focus on elution kinetics without additional experimental contexts limits the significance of the work, which may not meet the publication criteria for antibiotics.

Author Response: The authors would like to thank the reviewer for their critical appraisal of the manuscript, highlighting of potential limitations, and improvement suggestions provided. We have addressed the points raised by the reviewer at best possible in order to improve the manuscript’s quality and significance.

The authors do agree that one experimental setup only (i.e. antibiotic elution kinetics) poses a limitation to the herein presented work, yet it allowed to answer the stated research question, i.e. whether there are differences in antibiotic releasing properties of four antibiotic carriers used in clinical practice. The limitation of one methodology only has been added to the manuscript (see revised manuscript; Discussion, page 6, lines 202-205).

In addition, a novel figure is now included in the manuscript, depicting the real-life experimental setup of the study with the four 6-well plates containing the individually prepared antibiotic carriers (see revised manuscript; Methods, page 7, Figure 2).

Comment: Regarding Figure 1, it is recommended that the author employ a logarithmic scale for the time axis to enhance clarity and prevent the overlapping of time points, particularly at 0 and 8 hours.

Author Response: According to the reviewer’s valuable improvement suggestion, the elution kinetics of the four antibiotic carriers over time are now provided on a log-scale (see revised manuscript; Results, page 3, Figure 1).

Comment: In Section 4.3, pertaining to the in vitro elution assay, the methodology for sample analysis needs to be delineated clearly. The statement "a previously reported methodology" is insufficient. The method of analyzing vancomycin concentration is critical for the credibility of the manuscript, and readers would benefit from a detailed description of the analytical technique employed.

Author Response: Thank you for this comment. The methods section has been extended by now providing additional information on the study’s experimental setup with specific focus on preparation of the antibiotic carriers and the In-vitro elution assay (see revised manuscript; Methods, pages 7-9, lines 265-268, lines 274-279, lines 281-288, lines 292-298, lines 306-310, lines 314-323, lines 344-346). Furthermore, the phrase “a previously reported methodology” has been deleted (see revised manuscript; Methods, page 8, line 314). Additionally, as mentioned in the comment above, a figure showing the real-life experimental setup has been added to the manuscript (see revised manuscript; Methods, page 7, Figure 2).

Round 2

Reviewer 3 Report

Comments and Suggestions for Authors

The authors carefully addressed the points raised by the previous reviewers and, from my side, I don't have any additional observation. The manuscript, in my opinion, is ready for publication.